# Preliminary Report on Optical Coherence Tomography Angiography Biomarkers in Non-Responders and Responders to Intravitreal Anti-VEGF Injection for Diabetic Macular Oedema

**DOI:** 10.3390/diagnostics13101735

**Published:** 2023-05-13

**Authors:** Sanjana Chouhan, Rekha Priya Kalluri Bharat, Janani Surya, Sashwanthi Mohan, Janarthanam Jothi Balaji, V. K. Viekash, Vasudevan Lakshminarayanan, Rajiv Raman

**Affiliations:** 1Shri Bhagwan Mahavir Vitreoretinal Services, Sankara Nethralaya, Chennai 600006, India; sanjanachouhan1997@gmail.com (S.C.); rekhapriya26@gmail.com (R.P.K.B.); jananisurya92@gmail.com (J.S.); sashu23@gmail.com (S.M.); 2Department of Optometry, Medical Research Foundation, Chennai 600040, India; jothibalaji@gmail.com; 3Department of Instrumentation and Control Engineering, National Institute of Technology, Tiruchirappalli 620015, India; vkviekash.nitt@gmail.com; 4Theoretical and Experimental Epistemology Lab, School of Optometry and Vision Science, University of Waterloo, Waterloo, ON N2L 3G1, Canada; vengulak@uwaterloo.ca

**Keywords:** diabetic macular oedema (DME), optical coherence tomography angiography (OCTA), biomarkers, anti-vascular endothelial growth factor (VEGF) injection

## Abstract

Purpose: To identify optical coherence tomography angiography (OCTA) biomarkers in patients who were treated for diabetic macular oedema (DME) with intravitreal anti-vascular endothelial growth factor (VEGF) injections and compare the OCTA parameters between responders and non-responders. Methods: A retrospective cohort study of 61 eyes with DME who received at least one intravitreal anti-VEGF injection was included between July 2017 and October 2020. The subjects underwent a comprehensive eye examination followed by an OCTA examination before and after intravitreal anti-VEGF injection. Demographic data, visual acuity, and OCTA parameters were documented, and further analysis was performed pre- and post-intravitreal anti-VEGF injection.Results: Out of 61 eyes which underwent intravitreal anti-VEGF injection for diabetic macular oedema, 30 were responders (group 1) and 31 were non-responders (group 2). We found that the responders (group 1) had a higher vessel density in the outer ring that was statistically significant (*p* = 0.022), and higher perfusion density was noted in the outer ring (*p* = 0.012) and full ring (*p* = 0.044) at levels of the superficial capillary plexus (SCP). We also observed a lower vessel diameter index in the deep capillary plexus (DCP) in responders when compared to non-responders (*p* < 0.00). Conclusion: The evaluation of SCP in OCTA in addition to DCP can result in a better prediction of treatment response and early management in diabetic macular oedema.

## 1. Introduction

Diabetic macular oedema (DME) is a common cause of visual impairment in diabetes mellitus (DM) patients. DME is caused by the breakdown of the retinal blood barrier and leaking microaneurysms resulting in the accumulation of excess fluid and lipids in the macula. It is caused in secondary to multiple metabolic pathways triggered by hyperglycaemia [1]. VEGF is an important molecule responsible for the disruption of the blood–retinal barrier, and thus anti-VEGF agents are a primary treatment modality for DME. However, many patients show poor response despite multiple injections of anti-VEGF agents [2]. In a study by J.A. Wells et al., they reported that around 30% of patients are resistant to intravitreal anti-VEGF treatment [3]. Thus, it is important to identify factors in baseline examinations that can predict the treatment response to anti-VEGF agents.

OCTA is a new non-invasive technique that visualizes the retinal capillary plexuses layer by layer, providing valuable information on microvascular changes and the perfusion status of the retina, which thus helps to correlate functional and morphological data [4]. Previous studies have reported that eyes with DME had significantly lower vessel densities at the level of the deep capillary plexus (DCP) and a larger area of the foveal avascular zone (FAZ) when compared with eyes without DME [5]. Another study reported that the OCTA biomarkers of poor response were the enlargement of the FAZ area, the presence of more microaneurysms (MAs), and a lower vessel density (VD) of DCP [6]. A study focused on the OCTA parameters of microaneurysms and found that the accumulation of extracellular fluid at one year was strongly associated with hyper-reflective microaneurysms at baseline as compared to the hypo-reflective ones [7]. Busch C et al. defined the changes in the retinal vascular area as the percentage of the entire space occupied by large vessels and microvasculature in a particular region in SCP, DCP, and FAZ after intravitreal aflibercept in diabetic macular oedema, and concluded that a larger retinal vascular area, indicating preserved retinal perfusion, and the resolution of MAs were associated with better vision and an explanation of the DME after intravitreal aflibercept [8]. Thus, there seems to be evidence suggesting that changes in SCP, DCP, and FAZ may predict treatment response after intravitreal anti-VEGF treatment.

We analysed a retrospective series of subjects who had DME and were treated with intravitreal anti-VEGF. We observed the SCP, DCP, and FAZ characteristics on the OCTA machine to understand the differences in responders vs. non-responders and to identify biomarkers to predict treatment response to intravitreal anti-VEGF agents.

## 2. Materials and Methods

A retrospective cohort study of patients with DME (61 eyes) who received at least one intravitreal anti-VEGF injection was included from July 2017 to October 2020. The institutional ethics committee approved the research protocol and written informed consent was obtained from the participants. The study adhered to the tenets of the Declaration of Helsinki. All the subjects underwent a comprehensive eye examination followed by an OCTA examination before and after intravitreal anti-VEGF injection.

Inclusion criteria were patients with type 2 diabetes mellitus and diagnosed with DME, age >18 years; a very short period of follow-up patients were included. Patients with other retinal disorders, a history of vitreous surgery, or media opacities preventing high-quality imaging were not included.

Data collected included baseline demographics (gender, mean duration of diabetes, and medical history). Subjects underwent thorough ophthalmic exams, including slit lamp bio-microscopy and dilated fundus examination. Intraocular pressure (IOP) was measured with Goldman applanation tonometry. Best-corrected visual acuity (BCVA) was measured on a Snellen chart and expressed as the logMAR. The patients were classified into mild non-proliferative diabetic retinopathy (NPDR), moderate NPDR, severe NPDR and proliferative diabetic retinopathy (PDR) based on early treatment for diabetic retinopathy study (ETDRS) classification. We included 61 eyes of patients with DME who received one intravitreal anti-VEGF injection. The patients were divided into four quartiles for statistical analysis based on the difference in the reduction in central sub-field thickness (CST) pre- and post-intravitreal anti-VEGF infusion, as shown in Figure 1. Quartiles 1 and 2 were classified as group 1 (responder) and quartiles 3 and 4 were classified as group 2 (non-responder).

### 2.1. Acquisition of Images

All patients were imaged using Cirrus 5000 Angioplex (Carl Zeiss Meditec Inc., Dublin, CA, USA). Images of the macula were captured while centred on the fovea. The scanning area was captured on a 6 × 6 mm section (420 × 420 pixels) created from the intersection of the 420 vertical and the 420 horizontal B-scans. After the acquisition of images, the OCTA images were checked to ensure good focus, minimal saccades (identified by horizontal misalignment of vessel segments on en-face photos), and good centration. A signal strength of 7 and more was included for analysis.

### 2.2. OCTA Analysis

All scans were analysed using the automatically generated en-face OCTA images using the optical micro-angiography algorithm in the Cirrus OCTA software. The vessel density was calculated from SCP as well as DCP. Vessel density refers to the proportion of vessel area with blood flow over the total area measured and perfusion density refers to the total area of perfused vasculature per unit area in a measurement region. Vessel density is the proportion of the total area occupied by blood vessels in a specific region of interest within the OCTA image. It is expressed as a percentage and provides information on the number and distribution of blood vessels. Vessel density can be assessed in different retinal layers, such as the superficial capillary plexus, deep capillary plexus, and choriocapillaris, to evaluate the microvascular network. Perfusion density, on the other hand, is a measure of the total area occupied by perfused blood vessels in a specific region of interest within the OCTA image. It is also expressed as a percentage and provides information on the functionality of the blood vessels in terms of blood flow. Perfusion density takes into account both the presence of blood vessels and the actual blood flow within them, which is a more comprehensive assessment of retinal and choroidal perfusion. These SCP measurements were automatically obtained by the software that quantified the vessel density of a local tissue region according to the ETDRS subfields. The algorithm presents the values for the SCP, which is segmented from the internal limiting membrane to an estimated boundary of the inner plexiform layer. This inner plexiform layer boundary is calculated as 70% of the distance from the internal limiting membrane to an estimated limit of the outer plexiform layer, which is determined as being 110 μm above the retinal pigment epithelium boundary as automatically detected by the software. Figure 2 represents enface OCTA images of the SCP on the 6x6 mm scan. In the present study, we evaluated the vessel density and perfusion density in the central (mean value of subfield 1), inner ring (mean value of subfields 2 through 5), outer ring (mean value of subfields 6 through 9), and full ring (mean value of subfields 1 through 9) on the 6 × 6 mm scan pattern.

### 2.3. DCP Measurements

The FAZ was imaged using the commercially available spectral domain OCT (Cirrus 5000, Carl Zeiss Meditec Inc., Dublin, CA, USA) [9]. Angiography 6 × 6 mm program was used, and all images were aligned with the fovea as the centre point. All OCTA images were 8-bit grayscale images of 200 × 200 pixels corresponding to 6 mm × 6 mm (420 × 420 pixels). Each OCTA image was segmented by a clinician and used as the ground truth. This was compared with the new automated graphical user interface (GUI) method [10]. MATLAB R2020a App (Mathworks, Inc., Natick, MA, USA) was used to implement the algorithm. From these segmented images, DCP parameters and FAZ parameters were computed. As illustrated in Figure 3, the non-segmented image from which the FAZ region needs to be segmented is loaded first. The image description is then cropped off, and the FAZ region is plotted using the resulting image. The image is cropped around the region of interest to remove most of the vascular structures around the FAZ region for precise detection of FAZ boundaries and avoid false-positive segmentations.

Further, the Prewitt edge detector is employed to detect edges horizontally and vertically, thus distinguishing the boundary of the region of interest from its surroundings. Once the image’s edges have been recognised, we tend to reduce the noise caused by nearby vasculature, which might impair segmentation by causing false-positive detections. The closure procedure involved erosion, and dilation was used to remove the vascular structure. Additionally, this aids in the creation of a FAZ zone that is accurately divided. In this study, we used a line-shaped morphological feature to dilate the image at angles of 0°, 45°, and 90°. The angles chosen were evenly spaced, which aids in the preservation of the shape of FAZ.

Furthermore, a disk-shaped element was used to accomplish image closure and false-positive removal, which prevents the curvature of the FAZ boundary from being present. Because all false positives were discovered to be significantly smaller than the FAZs, we chose the largest. FAZ zones were identified and segmented using two types of marking: infill and outline segmentation.

### 2.4. FAZ Parametric Calculations

The algorithm calculates fifteen different parameters. Each parameter is explained below:Area (mm^2^): Area of FAZ, as shown in Figure 3, filled with green colour.Diameter (mm): The corresponding diameter of the circle with the same area.Major Axis (mm): Length (in pixels) of the ellipse’s major axis that has the same normalised second central moments as the FAZ region.Minor Axis (mm): Minor axis length of the same ellipse.Perimeter (mm): Length of the FAZ boundary.Eccentricity (mm): Measure of the departure of the region from circularity. For a circular region, eccentricity is essentially zero.Fmin (mm): Dimension of the most petite side of all rectangles that can completely contain FAZ, with every side of the rectangle tangent to the region.Fmax (mm): Dimension of the largest side of all rectangles that can completely contain FAZ, with every side of the rectangle tangent to the region.Inner circle radius (mm): The radius of the largest circle that can be inscribed in the FAZ region.Circumcircle radius (mm): The radius of the smallest circle circumscribing the FAZ region.Orientation of the major axis (degrees): The angle the ellipse makes with the horizontal axis.Blood vessel tortuosity (Dimensionless): Tortuosity describes the curve’s curvature based on twists and turns. Studies suggest that the thickness of the curve plays an important role in assessing tortuosity. The method indicated by Trucco et al. was used in this study to quantify tortuosity [11].Vessel diameter index (mm): The area occupied by the blood vessel from the binarised image over the total length of the blood vessel from the skeletonised image [12]. Qualitatively, it is the average diameter of all the vessels in the fundus image.Vessel avascular density (Dimensionless): Area occupied by vessels from the binarised image over the total area of the image. It is the density of vessels in the fundus image [12].Circularity index (Dimensionless): This dimension was calculated using both area and perimeter [13].

### 2.5. Statistical Analysis

Statistical analysis was performed using statistical software (SPSS for Windows Version 21; SPSS Science, Chicago, IL, USA). Data were entered in MS Excel 2016. The normality assumption was checked using the Shapiro–Wilk test. Table 1 provides the baseline characteristics of responders and non-responders to intravitreal anti-VEGF injection. Continuous variables are represented in terms of mean and standard deviation and were compared using Student’s *t*-test. Similarly, categorical data are presented as a number and percentage and were compared using the χ^2^ test. Percentage change in CST was calculated; if x is the baseline value, and y is the post-treatment value, then the percentage change was defined as (x − y)/x. Table 2 and Table 3 used the Student’s *t*-test to compare continuous variables among the two groups (eyes that showed good response to treatment with intravitreal anti-VEGF injection vs. eyes that showed poor response to treatment). Each set of the twelve comparison *p*-values was Bonferroni corrected (Bonferroni corrected *p* < 0.002).

## 3. Results

### 3.1. General Characteristics

A total of 61 eyes with DR and DME at different clinical stages were included in this study. Thirty eyes were included in group 1 (responder), and thirty-one eyes were included in group 2 (non-responder). The primary injection was bevacizumab (Avastin), ranibizumab (Accentrix), ranibizumab (Lucentis), and a bio-similar of ranibizumab (Razumab). The mean number of injections for responders was 1.666, and the mean number of injections for non-responders was 1.258.

Table 1 depicts the baseline and demographic features. There was no statistically significant difference in the baseline features between the two groups.

Out of the thirty eyes in group 1, twenty subjects were males, and ten subjects were females, with the mean age being 57.5 years. Thirty-one eyes were found to be non-responders (group 2), of which twenty-two were males and eight were females; the mean age was 60 years.

### 3.2. Association of SCP with Response to Treatment with Intravitreal Anti-VEGF Injection

We analysed the SCP features on OCTA in eyes with DME in both the study groups, as depicted in Table 2. We found a higher vessel density in the outer ring in group 1 that was statistically significant (*p* = 0.022). Similarly, a higher perfusion density was noted in the outer ring (*p* = 0.012) and full ring in group 1 (*p* = 0.044). There was no statistically significant difference between the two groups in the vessel density and perfusion parameters in the central and inner rings.

### 3.3. Association of DCP with Response to Treatment with Intravitreal Anti-VEGF Injection

The DCP parameters were also compared between the two groups (Table 2). We observed a statistically significant (*p* < 0.00) difference in the vessel diameter index between the two groups with a lower vessel diameter index in eyes that responded well to intravitreal anti-VEGF injections.

### 3.4. Association of Characteristics of FAZ with Response to Treatment with Intravitreal Anti-VEGF Injection

Table 3 shows the detailed analysis of the FAZ features in SCP and DCP between the two study groups. There was no statistical difference between the two groups in SCP and DCP.

Table 4 shows the percentage contribution of change in the responder and non-responder by each statistically significant SCP, DCP, and FAZ characteristic.

## 4. Discussion

We report the OCTA biomarkers in patients with DME treated with intravitreal anti-VEGF agents to predict response to treatment. There was no statistically significant difference in the baseline features between the responder and non-responder groups. We found the following features statistically significant in eyes with a good anatomical response to treatment with intravitreal anti-VEGF agents—vessel density in the outer ring in SCP, perfusion density in the full ring in SCP, and vessel diameter index in DCP.

OCTA biomarkers can play an important role in treating DME, as they can help identify macular ischemia, which can contribute to poor visual outcomes despite the resolution of macular oedema. Previous studies have looked at OCTA biomarkers that can predict worsening DR. A study showed that VD at the DCP correlated most strongly with the severity of DR [14]. In contrast, another study showed that VD in the SCP and not the DCP correlated with VA in DR [15]. One of the studies found that a lower VD in DCP was the only OCTA factor associated with the progression of NPDR and can be detected before the development of clinical retinopathy [16].

Only a few studies have looked at OCTA biomarkers for treatment response to intravitreal anti-VEGF injection. In a study by Hseih et al., a higher inner parafoveal VD in the SCP at baseline correlated most significantly with better visual outcomes after treatment with ranibizumab, thus serving as a predictor of visual improvement after anti-VEGF therapy. They adjusted for baseline BCVA, CRT, and EZ disruption. They showed that a lower parafoveal VD in the SCP correlated with a poorer visual improvement after treatment with ranibizumab, i.e., a poorer vascular perfusion resulted in lower vision. Even after the resolution of macular oedema, the visual acuity in these eyes with macular ischemia might not improve significantly. They also found that the outer parafoveal VD in the SCP could be predictive of visual improvement after treatment with ranibizumab [17]. This is similar to our study, where we found that eyes that responded well to intravitreal anti-VEGF injection had a higher VD in the outer ring and a higher perfusion density in the full ring in the SCP. Although our study did not consider visual outcomes, a good anatomical response is assumed to contribute to good visual outcomes.

In a study by Park et al., a decreased DCP/SCP flow ratio was observed in patients with DME that exhibited a poor response to treatment, i.e., the damage in DCP is more severe than that in SCP. Decreased VD in DCP is associated with worse visual acuity, suggesting that VD in DCP can reflect the degree of capillary loss in patients with poor vision in DME. Decreased VD and increasing VDI are associated with the worsening of the severity of DR [18]. In our study, a lower vessel diameter index in the DCP was noted in eyes that responded well to intravitreal anti-VEGF treatment. However, we did not find any significant difference in VD between responder and non-responder groups at levels of both SCP and DCP. Similarly, a study by Cheong et al. found no statistically significant difference between responders and non-responders in the VD in the SCP and DCP after three consecutive anti-VEGF injections [19]. However, some studies have reported an association between VD and response to treatment. Lee et al. found that DME patients with a poor response to anti-VEGF demonstrated a larger FAZ in the DCP [6]. Atallah et al. also evaluated the macular perfusion using OCTA in patients with treatment-naïve DME and moderate-to-severe NPDR. VD and FAZ were compared between diabetic eyes with DME, without DME, and healthy control.

Eyes with DME were found to have significantly lower VDs at the level of the DCP and a significantly larger FAZ compared to diabetic eyes without DME and control eyes. A larger FAZ was found to be associated with worse visual acuity [5]. Our study showed no difference in the FAZ area between responders and non-responders. The FAZ is typically circular in healthy individuals and becomes more acircular with greater changes in the FAZ perimeter, including breaks in the border and budding of tortuous capillaries into the FAZ in eyes with Diabetic Retinopathy [20]. The changes in FAZ are attributed to several pathological mechanisms, including capillary occlusion, endothelial dysfunction, and increased levels of VEGF leading to capillary abnormalities such as capillary dropout, vascular remodelling leading to increased tortuosity, and vessel loops [21,22,23].

A greater circularity index can indicate a better response to treatment since the normal FAZ boundaries are preserved in such patients. However, we did not find a statistically significant difference in our study. We found that the preservation of VD in the outer and inner rings and a good perfusion density were associated with reduced CST with treatment. An increase in VDI indicated dilated vessels in response to the capillary loss associated with poor response and increased CST after treatment. Our study shows that the evaluation of SCP and DCP is also important in determining treatment outcomes in patients with DME.

The limitations of our study include its retrospective nature and small sample size. Projection artifacts could interfere with the imaging quality of the deeper layers of the retina. Macular oedema could have affected the signal strength of the OCTA scans. Segmentation artifacts could also interfere with the results. Artifacts in the OCTA can make it a difficult task to evaluate DME. Our study was also not aimed at visual acuity outcomes. We also only looked at OCTA biomarkers after one intravitreal anti-VEGF injection. A better evaluation can be obtained after three loading doses with a longer duration of follow-up.

In conclusion, we found that parameters in both the SCP and DCP can help predict treatment response in DME. Further studies with larger sample sizes and longer duration of follow-up are required to adequately determine the biomarkers of OCTA, which can guide appropriate treatment with anti-VEGF injections.

## Figures and Tables

**Figure 1 diagnostics-13-01735-f001:**
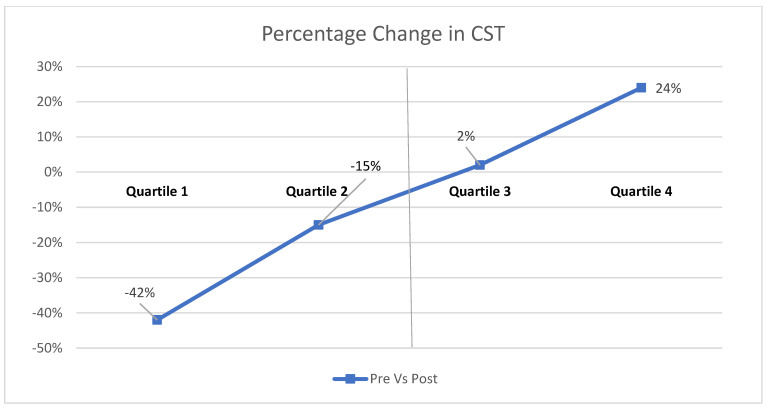
Graph showing the percentage change in CST pre- and post-intravitreal anti-VEGF injection divided into 4 quartiles. Quartiles 1 and 2 were classified as group 1 (responder) and quartiles 3 and 4 were group 2 (non-responder). CST: central subfield thickness.

**Figure 2 diagnostics-13-01735-f002:**
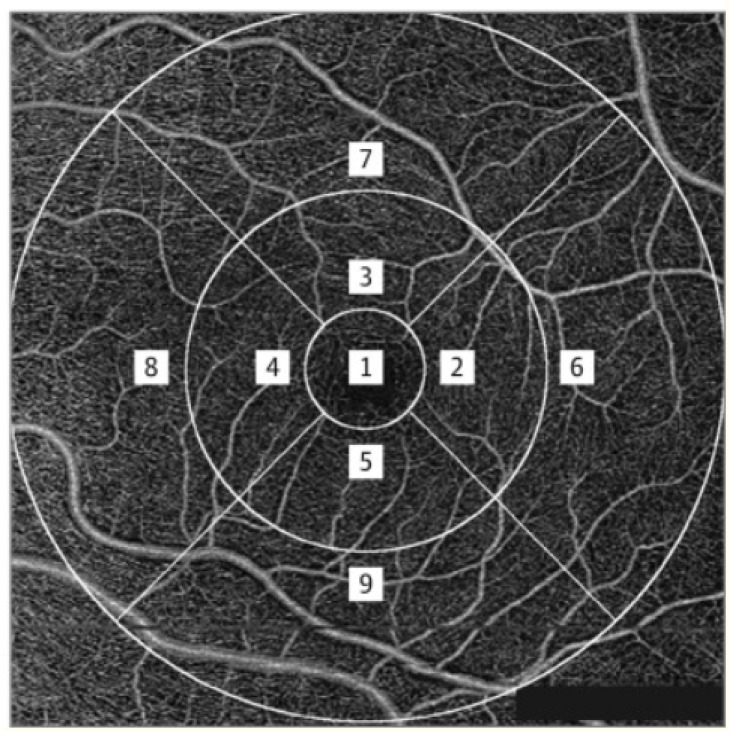
En-face image of the superficial retinal vasculature on OCTA overlaid with ETDRS grid. (1—Center, 2—Inner temporal, 3—Inner Superior, 4—Inner Nasal, 5—Inner inferior, 6—Outer temporal, 7—Outer superior, 8—Outer Nasal, 9—Outer inferior).

**Figure 3 diagnostics-13-01735-f003:**
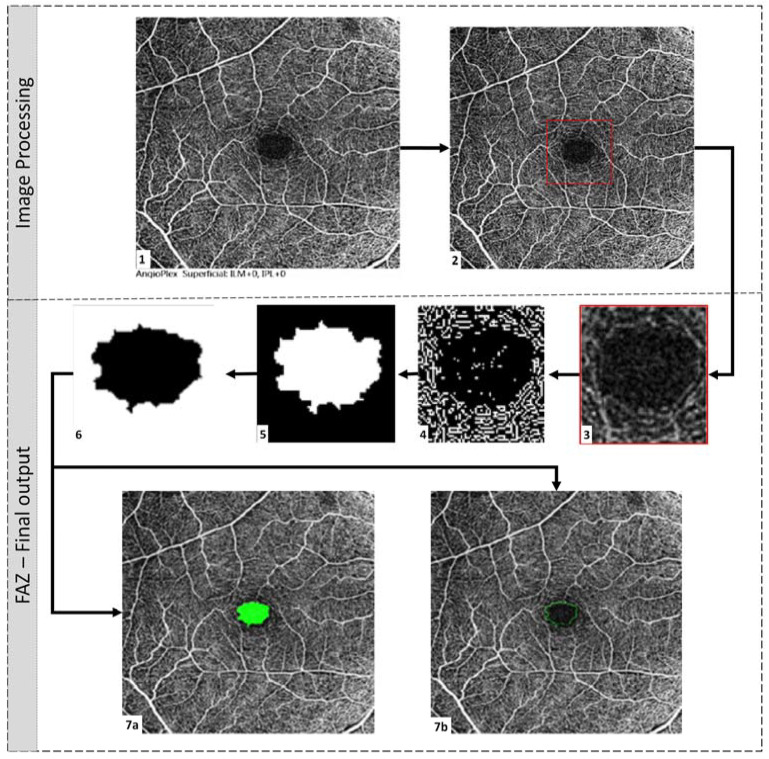
Methodology flow chart: Image processing: (**1**) Original non-marked image, (**2**) crop off the image description, (**3**) crop to the region of interest, (**4**) Prewitt edge detection, (**5**) image dilation, (**6**) image erosion and false positive removal, FAZ final output: (**7a**) infilled segmentation of detected FAZ region, (**7b**) outline segmentation of detected FAZ region.

**Table 1 diagnostics-13-01735-t001:** Baseline characteristics of responders and non-responders to intravitreal anti-VEGF injection.

Parameter	Responder	Non-Responder	*p*-Value
	(N = 30)	(N = 31)	
Age (years)	57.50 ± 6.50	60.10 ± 8.43	0.187
Gender			
Male	20 (66.7)	22 (73.3)	0.637
Female	10 (33.3)	8 (26.7)	
Duration of diabetes (in years)	13.83 ± 7.52	14.79 ± 9.48	0.669
CST (in microns)	494.0 ± 161.50	421.7 ± 121.68	0.042
BCVA (logMAR)	0.5 ± 0.3	0.6 ± 0.3	0.198

CST: Central subfield thickness, BCVA: best-corrected visual acuity.

**Table 2 diagnostics-13-01735-t002:** Difference in superficial and deep capillary plexus vessel characteristics of responders and non-responders to intravitreal anti-VEGF injection.

Parameter	Responder	Non-Responder	*p*-Value
	(N = 30)	(N = 31)	
SCP parameters			
Vessel density central ring	10.18 ± 4.41	9.69 ± 4.58	0.806
Vessel density inner ring	15.11 ± 2.92	14.28 ± 3.59	0.247
Vessel density outer ring	15.5 ± 2.2	14.55 ± 3.57	0.022
Vessel density full ring	15.29 ± 2.4	14.24 ± 3.51	0.077
Perfusion density central ring	22.89 ± 10.06	21.66 ± 10.87	0.756
Perfusion density inner ring	36.96 ± 7.81	34.1 ± 9.48	0.281
Perfusion density outer ring	38.79 ± 5.9	36.17 ± 9.6	0.012
Perfusion density full ring	37.93 ± 6.22	35.41 ± 9.32	0.044
Vessel tortuosity	1.34 ± 0.39	1.37 ± 0.28	0.306
Vessel avascular density	0.47 ± 0.14	0.43 ± 0.10	0.354
Vessel diameter index	28.85 ± 8.63	29.29 ± 6.67	0.464
DCP parameters			
Vessel tortuosity	1.38 ± 0.29	1.32 ± 0.27	0.376
Vessel avascular density	0.35 ± 0.10	0.33 ± 0.08	0.276
Vessel diameter index	27.01 ± 7.88	33.56 ± 16.95	0.000

SCP: Superficial capillary plexus, DCP: deep capillary plexus.

**Table 3 diagnostics-13-01735-t003:** Difference in superficial and deep capillary plexus foveal avascular zone characteristics of responders and non-responders to intravitreal anti-VEGF injection.

	SCP–FAZ Characteristics			DCP–FAZ Characteristics		
Parameter	Responder	Non Responder	*p*-Value	Responder	Non Responder	*p*-Value
	(N = 30)	(N = 31)		(N = 30)	(N = 31)	
Equiv. diameter	572.87 ± 231.42	582.96 ± 217.95	0.864	872.13 ± 331.82	909.08 ± 355.06	0.682
Major axis length	645.78 ± 289.11	663.61 ± 297.77	0.813	1036.76 ± 536.33	1134.09 ± 487.41	0.461
Minor axis length	497.63 ± 230.54	507.2 ± 208.75	0.866	734.01 ± 287.09	769.76 ± 296.48	0.641
Eccentricity	77.5 ± 43.29	85.87 ± 48.82	0.482	145.07 ± 101.36	160.25 ± 97.76	0.554
F minimum	510.89 ± 237.74	522.27 ± 214.88	0.845	704.66 ± 337.75	796.44 ± 300.9	0.267
F maximum	690.35 ± 273.28	717.16 ± 282.43	0.713	1141.52 ± 470.48	1184.95 ± 519.21	0.739
Inner circle radius	229.85 ± 96.91	234.35 ± 86.75	0.851	335.49 ± 133.96	356.26 ± 144.49	0.571
Circumference of circle radius	347.47 ± 137.61	362.22 ± 141.4	0.687	573.25 ± 234.8	594.07 ± 259.11	0.749
Orientation	−9.87 ± 37.51	−2.56 ± 47.17	0.514	−5.42 ± 34.84	5.16 ± 32.63	0.233
Area	0.3 ± 0.2	0.3 ± 0.24	0.955	0.68 ± 0.51	0.74 ± 0.53	0.643
Perimeter	1918.45 ± 872.91	1976.29 ± 842.98	0.793	2984.53 ± 1428.02	3317.63 ± 1405.74	0.362
Circularity	0.77 ± 0.24	0.79 ± 0.19	0.382	0.73 ± 0.20	0.75 ± 0.20	0.742

SCP–FAZ: Superficial capillary plexus–foveal avascular zone, DCP–FAZ: Deep capillary plexus –foveal avascular zone. Each set of the twelve comparison p-values is Bonferroni corrected (Bonferroni corrected *p* < 0.002).

**Table 4 diagnostics-13-01735-t004:** Percentage reduction in central sub-field thickness contributed by each of the significant factors.

Parameter	Responder	Non-Responder	Mean Difference	Percentage Decrease %	Status
Vessel density outer ring	15.5	14.55	0.95	−6.13	Decreased
Vessel density full ring	15.29	14.24	1.05	−6.87	Decreased
Perfusion density full ring	37.93	35.41	2.52	−6.64	Decreased
Foveal avascular zone circularity	0.51	0.49	0.02	−3.92	Decreased
Vessel diameter index in deep capillary plexus	26.75	34.26	−7.51	28.07	Increased

## Data Availability

The data was obtained from a tertiary eye care hospital, and the patient data will be maintained with confidentiality. The data will be shared upon request and can be provided only after the approval from the institutional review board.

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
