# Peer review of "Preliminary Report on Optical Coherence Tomography Angiography Biomarkers in Non-Responders and Responders to Intravitreal Anti-VEGF Injection for Diabetic Macular Oedema"

_diagnostics, 2023, doi:10.3390/diagnostics13101735_

Round 1

Reviewer 1 Report

Thank you for your trust and entrusting the role of a reviewer.  Diabetic retinopathy combined with macular edema is a growing problem. Treatment with anti-VEGF preparations is not always effective. Therefore, the attempt made in the study, aimed at determining the parameters allowing to determine the effectiveness of the therapy, is very up-to-date. The use of AOCT tests to assess the morphological parameters of the eyes also confirms the validity of the test. The statistical analysis used does not raise any doubts. Due to the short period of observation, the paper should contain the term "preliminary report" in the title. Please check the order of citations. Kind regards

Reviewer 2 Report

This is a really interesting research article concerning OCTA biomarkers in non-responders and responders to intravitreal anti-VEGF injection in patients affected by Diabetic Macular Edema.

This a well-written and well-organized paper, which was well carried out by the authors.

Only few points of this manuscript need to be clarified:

1) Did the authors tested the normality distribution of the data? The authors should specify this in the "statistical analysis" section.

2) The authors should specify NPDR and PDR at their first mention in the text.

3) The Tables 3 and 4 are not shown in the manuscipt, although they are mentioned in the text.

4) Minor English revision is required.

Reviewer 3 Report

Please see my comments attached, thanks.

In the manuscript, Raman et al. evaluated OCTA biomarkers of responders and non-responders to
intravitreal anti-VEGF injection for diabetic macular edema. OCTA images were captured, analyzed and
presented to show the differences between responders and non-responders. It is helpful for identifying
OCTA biomarkers which are significant parameters for DME treatment. However, the manuscript must
be improved before publication.

Detailed comments are listed below.

In the abstract and introduction part, full names do not have to show up whenever an
abbreviation is used.

Figure 1, what is range used to get retina thickness? What is the definition of sub-foveal region?

In the OCTA Analysis section, definitions of vessel density and perfusion density are the same?
Please clarify this.

Figure 3, figure captions are hard to read, please remake the figure and clarify it.

Table 3 and table 4 are missing in the manuscript, please complete and elaborate relevant
analysis results.

What does NPDR stand for?

In paragraph “In a study by Park et al., a decreased DCP/SCP flow ratio was observed...”, the
statement says, “Our study did not find any significant difference between the VD in responders
vs non-responders”, this is true for DCP only? Since eyes responded well to VEGF injection has
higher VD. Please clarify this.

The following statements are contradictory to each other, please clarify and provide supporting
evidence. “The FAZ is typically circular in healthy individuals and becomes more circular in DR”.
“FAZ is found to become more elongated with severe DR disease”.

Please add relevant references for the following statement. “The defects at the FAZ margin are
due to capillary dropout, increased tortuosity and vessel loops which can be due to increased
VEGF levels.

Round 2

Reviewer 3 Report

In this version, the authors addressed my previous comments and questions well. One minor question is listed below, thanks.

What is the essential difference between vessel density and perfusion density? Because for OCTA, to detect a vessel, blood cells have to flow through the vessel to generate the image contrast, which means the vessel has to be perfused. In this case, I don't see a difference between the two parameters. Can you please further clarify?
